# Rein Tension in Transitions and Halts during Equestrian Dressage Training

**DOI:** 10.3390/ani9100712

**Published:** 2019-09-23

**Authors:** Agneta Egenvall, Hilary M. Clayton, Marie Eisersiö, Lars Roepstorff, Anna Byström

**Affiliations:** 1Department of Clinical Sciences, Faculty of Veterinary Medicine and Animal Science, Swedish University of Agricultural Sciences, Box 7054, SE-750 07 Uppsala, Sweden; marie.eisersio@slu.se; 2Sport Horse Science, 3145 Sandhill Road, Mason, MI 48854, USA; claytonh@cvm.msu.edu; 3Department of Anatomy, Physiology and Biochemistry, Faculty of Veterinary Medicine and Animal Science, Swedish University of Agricultural Sciences, Box 7046, SE-750 07 Uppsala, Sweden; lars.roepstorff@slu.se (L.R.); anna.bystrom@slu.se (A.B.)

**Keywords:** transition, horse, gait, rein tension, dressage, kinematics

## Abstract

**Simple Summary:**

In the equestrian dressage discipline, the transitions (changes) between gaits and into halts, occur often in riding sessions. Rein tension before, during and after the transitions between gaits, and the transitions into halts were studied. The vertical motion data for the horse’s head and croup, and rein tension data were collected from six professional riders, each riding three of their own horses during normal training sessions. The horse training levels varied from basic to advanced. The activities during the sessions were categorised into gaits, transitions between gaits and into halts based on video evaluation. The transitions were categorised according to whether they had intermediate steps that were not characteristic of the preceding or the following gait. The rein tension just before the transition was strongly related to rein tension during the transitions. There was slightly lower tension during the upward transitions than during the downward transitions. There was no difference in rein tension depending on whether intermediate steps were present or not. The left rein tension was generally lower than the right rein tension. The rein tension associated with the transitions and halts varied substantially between riders and also the horses. This information is useful for trainers seeking to understand the rein tension patterns associated with transitions.

**Abstract:**

In dressage, the performance of transitions between gaits and halts is an integral part of riding sessions. The study aimed to evaluate rein tension before, during and after the transitions between different gaits and the transitions into halts. The kinematic (inertial measurement units) data for the head and croup, and rein tension data, were collected (128 Hz) from six professional riders each riding three of their own horses, training levels varying from basic to advanced, during normal training sessions. The activities were categorised into gaits, halts and transitions based on video evaluation. The transitions were categorised as without (type 1) or with (type 2) intermediate steps that are not normally present in the gaits preceding or following the transition. The differences in the median rein tension before/during/after transitions, between the types and left/right reins were analysed in mixed models. The rein tension just before the transition was the strongest determinant of tension during the transition. The rein tension was slightly lower during upward transitions compared to downward transitions, reflecting the pattern of the preceding gait. Type 1 and 2 downward transitions were not different regarding rein tension. The left rein tension was lower than right rein tension. The rein tension associated with the transitions and halts varied substantially between riders and horses. The generally strong association of the gaits and their inherent biomechanics with rein tension should be taken into account when riding transitions and halts.

## 1. Introduction

The horse’s three most common gaits are walk, trot and canter. When moving freely without being constrained by a rider, transitions between these gaits are performed for energetic economy [1], to reduce limb forces [2] and/or to preserve gait stability [3]. When carrying a rider, the horse is expected to change gait in response to an aid from the rider rather than in response to a biomechanical or physiological cue. One of the rider’s tasks when educating the horse is to teach the horse to respond correctly to cues for adjusting speed. This includes both to change speed within a gait, and to transition between gaits, which requires a change in footfall sequence.

The transitions between gaits during riding are challenging for both horse and rider. When performing transitions between gaits, the rider aims for the horse to change directly from the footfall sequence of one gait to that of the other gait, which is called a direct or type 1 transition. If there are intervening steps that produce limb support sequences not typical of either gait, it is an indirect or type 2 transition [4,5]. The horse should move in a balanced and sustainable way, with a steady head carriage, before, during and after the transition which is challenging because gait perturbations are an integral part of the transition. During a transition between gaits, the gait pattern changes more or less abruptly and the rider needs to follow the horse’s movements while continuing to interact with the horse via the reins as well as the legs/seat. The rein tension varies in magnitude throughout the stride with a gait-specific pattern [6,7,8] and it is substantially affected by both the horse and rider [9]. The gait-specific patterns reflect the accelerations and decelerations of the horse’s body and the associated head and neck oscillations that are synchronized with the footfall sequence in each gait [7,8,10]. Excessive head movements during a transition may reflect that the horse is struggling with balance, or resisting the rider’s aids [11].

The upward and downward transitions refer to transitions that involve an increase or decrease in speed within or between gaits. It is often recommended that the rider should prepare the horse with a half-halt before performing a transition [12]. In a half-halt, the horse should step further forward underneath its body and carry more weight on the hind limbs, without losing impulsion [12]. The aids for upward transitions usually involve a combination of cues from the rider’s seat and legs, possibly accompanied by a light rein aid. The primary aid for decelerating or a downward transition is an increase in rein tension, both for the preparatory half-halt and for the transition [12]. The increase in rein tension is combined with aids from the rider’s seat and legs to indicate which gait should be performed after the transition. Through the learning principle of negative reinforcement, the horse learns the meaning of different pressures in the mouth [13]. Negative reinforcement is a form of operant conditioning in which a stimulus is withdrawn in response to performance of the correct behaviour. If pressure is consistently released as a consequence of correct behaviour, there is an increased likelihood that the same behaviour will be performed again. For example, the horse learns, through trial and error, that responding to rein tension signals by decelerating, performing a downward transition or halting is followed by a reduction in rein tension [13].

As the horse’s training under a rider progresses, the promptness of the transitions improves [4,5,14], but the actual transition duration may be prolonged [15]. In the transitions to halt, it was found that the more steps the horse took before stopping, the more the mean rein tension used to achieve the halt increased [16]. Therefore, a prompt response from the horse to a rein tension signal, and a prompt release from the rider, are key factors for maintaining rein tension at low magnitudes.

The aim of this study was to examine professional riders riding their own horses during a normal training session to determine the type and distribution of gait transitions and halts. The synchronised measurements of rein tension and kinematics of the horse’s head and croup, before, during and after the transitions were analysed to assess the patterns in these variables associated with the transition category (gait before and after) and type (type 1, type 2), including the variation between and within riders and horses. This study hypothesised that rein tension would differ between upwards and downwards transitions. It was expected that the downwards transitions would be associated with higher rein tension than the upwards transitions and that transitions involving gaits in which the horse’s head has a large range of motion (ROM) would be associated with high rein tension. It was hypothesised that rein tension would be affected by the transition type, e.g., that direct transitions would be associated with lower rein tension than those with intermediate steps.

## 2. Materials and Methods

The rein tension data were collected from 6 professional riders (mean + STD height 173 + 6 cm and weight 65.5 + 10 kg) each riding 3 horses (*n* = 18) that they had been training for a period of time (mean: 7 years; range: 1 month to 22 years). The educational level of the horses was reported by the riders as: Young horse (*n* = 3), basic (*n* = 6), medium (*n* = 5) and advanced (*n* = 4). The advanced horses had competed at Prix St. George, Intermédiaire or Grand Prix level. The basic horses had entered low-level competitions only and medium horses were in between. Young horses had been ridden for less than a year and had not competed. Based upon the evaluation of videos recorded during the study, all horses were judged by a veterinarian to be free from lameness. Each horse wore its own well-fitting saddle and bridle with a snaffle bit. Ten of the snaffles had three parts, one was straight and made of rubber, and seven had two parts. Two of the ten 3-part snaffles had fixed rings, one had a small port and two of the 2-part snaffles were full-cheek. Further information on the horses and riders can be found in Eisersiö et al. [17]. When asked, five riders stated they were right-handed and one left-handed. To assess horse sidedness patterns, the riders were asked the following question: “To which side does your horse bend more easily, left or right?” The riders reported that five horses bent more easily to the left, 11 horses bent more easily to the right and one horse was equally easy to bend to the left and right. The riders provided written consent regarding their participation, animal or human ethical approval was, according to Swedish legislation [18,19], not necessary to conduct the study.

The data collection took place at each horse’s current stable yard in an indoor (*n* = 3 riders) or outdoor (*n* = 3 riders), synthetic sand-fibre (*n* = 3) or sand (*n* = 3), riding arena, depending on the weather conditions, and with no other horses in the arena. Prior to mounting by the rider, custom-made rein tension meters with recording frequency 128 Hz, measuring range 0–500 N, resolution 0.11 N [19], were fitted onto the leather reins on the left and right sides of the bridle. A cable from each tension meter ran forwards along the rein and up the cheek piece of the bridle, to an Inertial Measurement Unit (IMU, x-io Technologies Limited, UK) that logged the rein tension data. The IMU was fastened to the browband of the bridle using hook and loop fasteners. The rein tension meters had been tested for stability and repeatability in a tensile testing machine and were calibrated before starting the riding sessions of each rider by suspension of 13 known weights (0 to 20 kg), added in ascending order.

A second IMU (128 Hz) was taped over the horse’s dorsal midline at the level of the first sacral vertebra. Video recordings (Canon Legria HF200, 25 Hz) of the entire riding session were made from the middle of one of the long sides of the arena. After the rider had mounted at the start of the session and again before dismounting in the end, the rein tension meter was synchronized with the video recordings by applying tension to the right tension meter 5 times while counting out aloud in front of the camera, then repeating the process a second time. The two IMUs were synchronised with each other before and after each session, by placing them on a board and then tapping the board lightly. At start, it took approximately 10 min to fit the equipment onto the horse and synchronize the equipment.

The riders were asked to perform a normal flatwork/dressage training session with each horse, including periods of walk, trot and canter, but with no instructions on figures or exercises. The whole riding arena was used for the exercises and the duration of the riding session was determined by the rider. The duration varied from 22 to 44 min with median 31 min.

Video scrutiny was used to identify the transitions and halts, as well as the gaits and exercises throughout the sessions [17]. This included coding of whether riders rode on long/slack reins or with short reins, which was defined as the reins being held so that the rider has contact with the horse’s mouth. Based on the visible footfall sequences, walk, trot, the two canter leads, rein back, piaffe, and sequences of unidentified gaits/movements were located on the videos by one author (ME) and reviewed by another author (AE). The transitions were categorised according to the gaits that preceded and followed each transition. Based on Argue and Clayton [4,5], the transitions were categorised according to the limb support sequence as type 1 (all limb support sequences during the transition were typical of the gaits before or after the transition) or type 2 (included limb support sequences not typical of the gaits before or after the transition) (Appendix A). Only the frames in which horses moved between gaits were identified as belonging to the transitions. As an example, a walk-trot type 1 transition started from the first disruption of the walk sequence and ended at the first stance diagonal of trot. The halts were identified and categorised according to the preceding gait or motion, e.g., trot-halt transition. The simple standstills when the rider stopped briefly on long reins (e.g., to adjust the stirrups) were not included other than in an overall event count. 

The IMUs on the head and sacrum provided acceleration and gyroscopic data, and calculated Euler angles. The logged rein tension and IMU data were managed in Matlab (Matlab version 2016b, The MathWorks^®^ Inc., Natick, MA, USA). The nose angle range of motion (ROM) per transition, defined as the maximal minus the minimal nose angle, was determined from Euler angles from the head unit attached to the browband. Positive head pitch rotation was defined as a clockwise rotation around the transverse axis when viewed from the left side, i.e., a more positive value indicated the nose was more forward and the bridge of the nose became more horizontal. Positive croup roll rotation was defined as a clockwise rotation around the longitudinal body axis when viewed from behind, i.e., a positive value indicated that the right tuber coxae was lower than the left. The poll and croup acceleration signals were double integrated and high-pass filtered with cut-off values from 0.2 to 0.5 Hz and order 1. The raw data, including the rein tension, heights of poll and croup, head pitch and croup roll, were plotted versus the time of the session.

All the transitions between gaits were counted, but only those performed with short reins were analysed further. The duration of each transition was derived from its start and finish times during the riding session. Type 1 transitions that lasted longer than 1 s and type 2 transitions that lasted longer than 1.5 s were eliminated. 

For each transition, the descriptive variables were calculated for tension in the left and right reins. The median rein tensions (each rein separately and a mean of median left and median right rein tension) and ROM nose angles were determined per gait during the sessions, and for the period starting 1 s before the transition, during the transition and for the period ending 1 s after the transition. In principle, the same data were derived for halts, the data for 1 s that preceded the horse becoming stationary, and during the halt until the horse moved off. Since a halt was often followed by activities unrelated to training, such as adjusting the girth or a pause on loose reins, the data after the halts were not included in the analysis.

The data are found in Appendix A. The number of (short-rein between-gaits) transitions and halts of different categories and types were tabulated. The median left and right rein tension during the transitions were visualised in boxplots by the rider, horse and transition category. The descriptive statistics were tabulated for the variables used for purely descriptive purposes and for those used in modelling (both parametric and non-parametric because of varying normality for presented variables).

The explorative models, models (a), were performed in order to study the associations between selected variables in the dataset and the median rein tension during the transitions, for the left and right reins combined, and for the right rein which had more complete data. The dependent rein tension data were logarithm transformed guided by a Box-Cox transformation. The independent variables selected were: Transition type, right or left/right combined median rein tension during 1 s before the transition, duration of the transition, the percent of the session when the transition occurred, and ROM of the nose angle during the transition and during 1 s before the transition. The linearity of variables versus the outcome variables was investigated by plotting the independent variables versus the dependent variables. For the right, or left/right combined, median rein tension during 1 s before the transition, the relationship was nonlinear and this variable was modelled adding a square of the variable (right or left/right combined median rein tension). The random effects were rider and horse within rider. Each independent variable was tested upon the random effects in a single fixed effect model (i.e., one independent variable at a time).

Secondly, to study the association between the median rein tension and sequence in the transition (before, during, after), the transition types and left/right rein, a more specific model (b) was developed. The dependent variable was the median left and right rein tension. The rein tension data were transformed to the 0.25 root guided by a Box-Cox transformation. Transition type (Appendix A), stage of transition (before, during, after) and rein (left/right) were included as fixed effects. The random effects were the transition number, rider and horse nested within the rider. Two-way interactions between the fixed effects (ignoring the rein and transition type interaction), and between the fixed and random effects were considered. Guided by Akaikes information criterion and type III *p*-values < 0.05, the models were reduced backwards. A corresponding model was developed for the outcome median rein tension before and during the halt (model c), including data on halts from walk or trot and including gait before the halt, stage of the halt (before and during) and rein (left/right) as fixed effects.

Thirdly, in order to compare the median rein tension during 1 s before the transition to the rein tension used for the same gait during the whole sessions when riding on short reins, model (d) was developed. The outcome variable was the difference between the median rein tension before the transition and the median rein tension for that gait during the entire session. The data were transformed using the reciprocal inverse using a Box-Cox transformation. The fixed effects were the transition type and rein. All other procedures were done as in model b.

The covariance structure was set to identity in all models (proc mixed, SAS Institute Inc., Cary, NC, USA). The *p*-value limit was <0.05. The model estimates are presented for the continuous variables and the least square means are presented for classification variables. The latter are accompanied by back-transformed results, with 95% confidence intervals (95% CI) when allowed by the transformation. For transition type, predetermined pairwise comparisons were selected.
Types 1 and 2 were only compared between the same stage of transition (e.g., between type 1 and type 2 for the transition from canter to trot during the transition, applicable to models a and b).Upwards and downwards transitions between the same pair of gaits and of the same type were compared (e.g., type 1 transitions from canter to trot, and trot to canter, models a and b).The stages of the transitions were only compared within type (model b).

## 3. Results

### 3.1. General Data

A total of 538 transitions (14–53 per horse) were categorised based on footfall sequences from the videos. Of these, 527 were made with short reins. The number of short-rein transitions, within the duration selected (Type 1 transitions that lasted longer than 1 s and type 2 transitions that lasted longer than 1.5 s were eliminated) that also had the data on rein tension (before, during, after) was 496 transitions with the right rein data (10–51 per horse) and 427 transitions with the data for both reins (1–46 per horse). The smaller number of transitions recorded for both reins was due to data loss for the left rein. The duration of the transitions was 0.2–1.4 s (median 0.5 s). The distribution of the 496 transitions among the categories and types are shown in Table 1.

There were 160 halts identified (2–35 per horse), 111 of which were performed with short reins (1 to 28 per horse). The halts performed on short reins and with the rein tension data before and during the halts were 108 for the right rein and 100 for both reins. The distribution over categories is shown in Table 1. The duration of the 108 halts varied from 0.2 to 13 s, median 3.0 s.

### 3.2. Graphical Presentation

The raw rein tension and kinematic data for the transitions (Figure 1 and Appendix A) and halts (Figure 2 and Appendix A) are presented. The supplements show all the transitions/halts on short reins for three horses. The upper curves demonstrate left and right rein tension, colour-coded by gait (see legend Figure 1). The dimensionless lower curves show temporal associations with the head and croup height, nose angle and croup roll in the transitions (Appendix A is made to aid in understanding of the time-series graphs).

Figure 1 shows six consecutive transitions from one horse. The rein tension magnitudes and patterns show similarities, especially for the two trot-walk type 1 transitions in which the direction of the croup roll shows that the last step of the trot was a left hind diagonal and the next step was the right hind diagonal which was the first step of the walk. Both were thus type 1 transitions initiated from the same diagonal (Appendix A).

The six graphs in Figure 1 each represent a single transition as shown in the bold-line red/blue colour and if adjacent transitions occurred, they are represented by gaps in the data. In Figure 1, the transitions represented in graphs 1 and 2 occur close together during the session. The horse took 2–3 strides of the trot between a left lead canter and a walk.

### 3.3. Magnitude and Variation of Rein Tension and Other Variables

The between-rider and between-horse variation in rein tension during the transitions are substantial, as illustrated in Appendix A. These boxplots demonstrate the left and right median rein tension during the transitions, by horse and transition category, and also show the median rein tension per gait during the whole sessions. In order to appreciate the magnitude and variation in the variables studied, Appendix A tabulates statistics for a number of variables evacuated from the data, including data on minimum and maximum rein tension applied during the transitions and halts and data distributions for variables used for modelling.

### 3.4. Modeling

#### 3.4.1. Exploratory Models

The results of the exploratory models, models (a), for which *p* < 0.1 are shown in Table 2. There was a strong association between rein tension before and rein tension during the transitions (Table 2) and this association was linear on the untransformed scale (Figure 3). Thus, given the logarithm transformed outcome, squared before rein tension was needed in the models (Table 2). The transition type was significant with a type III *p*-value of <0.0001 in the models for the right rein, and the left and right rein combined. There were significant differences between the same-gait trot-canter types 1 (22–23 N) and 2 (16 N), in both models *p* = 0.04. There were significant differences between opposite gaits (e.g., upwards and downwards transitions of type 1, Appendix A). An unsteady head, represented by changes in the nose angle in the period of 1 s before the transition, was associated with an increase in rein tension during the transitions (Table 2). The transitions that occurred later in the session were borderline significant for an increase in right rein tension. For the right rein as outcome, the tested variables that had a *p*-value > 0.10 were the nose angle during the transition (*p* = 0.66), and the duration of the transition (*p* = 0.70), and for both reins nose angle during the transition (*p* = 0.41), the duration of transition (*p* = 0.47) and the percentage of session (*p* = 0.20).

#### 3.4.2. Stage of Transition Model

The data distributions before and after transformation are presented (Appendix A). The model results for the stage of transition model (b) are found in Table 3 (*n* data points = 2769, *n* transitions = 496). The reduced model includes the fixed effects of transition type (type III; *p* < 0.0001), stage of the transition (*p* = 0.25), the rein (*p* = 0.02) and the interaction between the transition type*stage of transition (*p* < 0.0001). The variation estimated from the random effects was: Transition number 25%, rider 11%, horse within rider 18%, horse*transition type 14%, horse*rein 1.5%, horse*stage of transition 0.2% and the residual variation was 30%. The least square means and significant comparisons are shown in Table 3. The tension in the left rein was lower (22 N) compared to the right rein tension (25 N), controlling for other variables in the model (*p* = 0.02). Comparing within types and between stages of the transition, there were multiple significances. Comparing upward and downward transitions between the same pair of gaits (within type), there were three significant differences. These latter comparisons revealed that rein tension was lower in the upward compared to the downward transitions (Table 3).

#### 3.4.3. Rein Tension in Halt Model

Model (c) with the outcome median rein tension before and during halts was made on 0.25 root transformed data (*n* data points = 316, *n* halts = 84). The model includes the fixed effects of halt (from walk or from trot) and stage of the transition (before/during). The variation estimated from the random effects was: Transition number 59%, rider 9%, horse within rider 16% and the residual variation 15%. The halts from the trot were associated with a higher median rein tension (back-transformed least square mean 16.2 N) than those from the walk (9.4 N, Walds *p* = 0.0006). The median rein tension was higher before (13.4 N) than during the halt (11.5 N, Walds *p* < 0.0001). The interaction between the stage of the halt and gait before the halt was non-significant (*p* = 0.44), and the left and right rein variable left the model at *p* = 0.053.

#### 3.4.4. Rein Tension before Transition Model

Model (d) elucidates the differences in rein tension before the transitions, compared to median rein tension during the whole session for the same gait and rein. The transition type remained significant (type III *p*-value < 0.0001). The back-transformed least square means (Table 4) are all positive, i.e., the tension applied during the period of 1 s before the transition was generally larger than the median rein tension (in the data only 25% of the 913 differences were negative, including both reins and all transition types).

## 4. Discussion

This is the first study that both categorised the transitions and assessed rein tension associated with those transitions. It is important to acknowledge the observational nature of the study. The riders were instructed to do what they would normally do during a riding session, and they were not informed of the focus on the transitions. This is reflected in the considerable variation between the horses and riders in the number of transitions and halts made, the types of transitions, and in rein tension during the transitions. It can be expected that the character of the transitions during the warm-up, the main working phase, and the cool down may differ and that the transitions may be affected by the horse’s current level of training and by the exercises performed between transitions, something that could be studied in the future. The goal of making a transition will vary, from simply changing gaits, to enhancing the quality of the gaits, to perfecting the performance of the transitions. The natural set of the study is illustrated by the two graphs in the upper row of Figure 1, where two transitions are very close in time. The rider may have intended to do a canter to walk transition that instead developed into two transitions, one from canter to trot and one from the trot to walk. This may have been a consequence of the horse and/or rider-related factors. The indirect transitions are often not desirable at a competition, where a judge would penalize steps of an intermediate gait during a transition.

The most common transitions were between the walk and trot, followed by those between the trot and canter. Overall, type 1 transitions represented the highest proportion (85% of all 253 upward transitions, and 88% of all 243 downward transitions, Table 1). The trot to canter transitions had the highest proportion of type 2 transitions. The predominance of type 1 transitions may be related to the high level of training of the riders and horses or it may be that horses prefer the more simplistic transitions. Most halts were made from walk, the halts from walk likely being easier to command and thus perform (with perfection) compared to those from gaits with higher speed or momentum. The duration of the transitions in the current study was much smaller (5–95 percentiles 0.5–0.9 s) than those defined by changes in the horse’s acceleration [15], reported as several seconds. Our definition of start and end were marked by a change in footfall sequence, as seen from the video.

The exploratory model of the median rein tension during the transition identified ROM nose angle (Table 2) before the transition as a significant variable. Moving the head during the period of 1 s before the transition was associated with higher rein tension during the transition. From previous studies, it has been seen that the horse’s gait is a strong predictor of the magnitude and pattern of rein tension [6,9,10,11,12,13,14,15,16,17,18,19,20,21,22] due to the gait-specific, cyclic acceleration patterns of the horse’s head. The rein tension generally increases with speed. It is higher in trot than walk, higher in canter than trot, and higher in the extension than the collection in trot and canter [9]. It was therefore suspected that the significance of the head movements was driven by which gait preceded the transition. The analyses were further stratified by gait before the transition. When trot preceded the transition (but not walk or canter), the estimate for ROM nose angle was significant (estimate 0.008, SE 003, *p*-value = 0.006, *n* = 183 transitions, left and right rein tension data combined, Appendix A) with an increase in ROM nose angle predicting an increased median rein tension during the transition. The changes in the head angle may indicate that the horse is actively resisting the action of the bit as the rider gives a preparatory half-halt. The activation of the rider’s biceps brachii in early stance and triceps brachii in late stance has been shown to stabilize the rider’s hands so they can maintain contact with the bit during trotting [21]. During a transition, it is more difficult to stabilise the contact due to the horse’s movements being less predictable and this requires the rider to respond via a feedback mechanism. The fact that all riders were accustomed to the horses and the way those horses perform the transitions likely benefitted the consistency of their responses. 

The time during the session when the transition occurred was positively associated, *p* = 0.05 (Table 2), with rein tension during the transition in one of the exploratory models, suggesting that the transitions later in the sessions had higher rein tension. Stratifying the analysis, there was a significant result only for walk to trot transitions (estimate 0.006, SE 002, *p*-value = 0.02, *n* = 128 transitions, left and right rein tension data combined, Appendix A). Perhaps, the walk-trot transitions later during the session were done with more emphasis on responsiveness, including relatively shorter reins, or the horses had habituated [13], and relied more on bit support for maintaining balance, and a higher rein tension resulted.

Our presumption that the more parsimonious type 1 transitions would have lower rein tension than type 2, with more steps between the gaits, was partly supported by Warren-Smith et al. [16] who found that rein tension increased with the number of steps taken before coming to a halt. This study found a significant difference between rein tension during the upward trot-canter transitions type 1 (23 N) and 2 (16 N) (Appendix A). Hence in this study, only one out of four between-type comparisons in the exploratory models (model a) yielded a significant result. Significant results for similar comparisons were not found in the transition stage model, nor was it found that the duration of the transition affected rein tension in the transition. The authors conclude that the transition type does not seem to be a strong determinant of rein tension during the transitions. It could be that the tension at the transition is more related to the promptness in initiating the transition rather than whether the horse performs it with or without intermediate steps. However, it is suggested that larger observational or experimental studies including more data for each transition type, and including riders and horses performing in different disciplines, are needed in order to meet the substantial between-individual variation. This is feasible if electronic equipment registering high quality kinematic and rein tension data are used by many riders on a regular basis. The advantage of such studies is a representative overview of the situation for the horses in the general population. More variables describing the rein tension signals should be studied. For example, if type 2 transitions take much longer time to complete than type 1 transitions, then the rein tension impulse (the area under the rein tension curve) can be greater, but this was not studied here.

The rein tension before the transition (Table 2, Figure 3) was the most significant and strong predictor of rein tension during the transition in the exploratory models. Therefore, the modeling continued with contrasting rein tension before, during and after a transition, as well as comparing to median rein tension over the whole session. When rein tension within 1 s before, during, and within 1 s after the downward and upward transitions between the gaits walk, the trot and canter was assessed, it was found that rein tension gradually decreased when transitioning from a faster gait to a slower gait (downward transition) and that rein tension gradually increased when going from a slower gait to a faster gait (upward transition, both Table 3, differences between opposite gait pairs). This suggests that the biomechanics of the gaits are influencing rein tension during the transitions.

Typically, the horse is first taught that pressure applied in its mouth from the bit means slow down [12]. As training progresses, the young horse learns to respond to many other rein tension signals, which may be given in combination or in close proximity to signals applied from the legs and seat of the rider [12]. The tension in one or both reins can thus also mean turn, turn your head and/or bend your neck, yield at the poll or lighten the contact with the reins, depending on the context [12]. In dressage, it is popular to ride with support, i.e., to ride with a constant tension on the reins that the horse habituates to and is used as a basic contact on the reins [10]. The benefit of having a baseline tension is that it allows more subtle communication between the horse and rider than would be the case if contact between the rider’s hand and the horse’s mouth had to be re-established each time a rein aid is given. In a previous study, we reasoned that the average values of the minimum rein tension per stride could be used as a measure of the basic support tension [10,22]. The magnitude of the baseline rein tension seems to be specific to each horse, rider and gait (Appendix A). Superimposed on the baseline tension is a gait-specific pattern of rein tension changes [6,7,8,10]. The findings presented here suggest that rein tension before, during and after transitions is influenced by the gaits between which the transitions are made.

This study demonstrates that kinematic measurements from two midline IMUs provided considerable information beyond rein tension alone. With regards to gait mechanics, the vertical motion of the axial body segments, as well as pelvic rotations, reflect specific events within each gait. For example in the first graph of Figure 1, for the transition between right canter and trot, both the head and croup movement are regular and obvious in canter prior to the transition to trot, while after the transition, the head motion is less distinct and of lower range, which is characteristic for the trot (there is often a degree of interference with the sinusoidal motion of the head when the rider has contact with the reins). For both the canter and the trot, the lowest head position (highest acceleration) coincides with midstance [23]. Type 1 transitions often involve uniting (e.g., from walk to trot) or breaking up (from trot to walk) a diagonal which can be identified from the data (Appendix A). For example, in the second graph of Figure 1, the trot-walk transition is initiated at right hind limb midstance. The preference of horses and riders to initiate a transition on the left or right diagonal could be studied for further elucidation of sidedness during the transitions, similar to the studies of forelimb protraction preference when eating from the ground [24,25].

Controlling for the transition type, left rein tension was lower than right rein tension (Table 4). Previous studies have described differences between left and right reins [9,26,27]. By analysing the range of rein tension, the left rein tension was less than the right rein tension in a group of eight riders riding 24 horses [9], the riders in the present study being a subset of the riders in that study. By analysing the differences in the area under the rein tension curve (the impulse) on a stride by stride basis in 17 horses, 11 horses had a significantly higher tension in the right rein, four horses in the left rein and in two horses left-right rein differences were non-significant [27]. This may be connected to consistent kinematic sidedness patterns in individual horses or riders. Human and horse laterality are key variables to consider when performing such evaluations [28]. However, because of the relatively low power in the current study, the rider or horse sidedness were not analysed.

Not surprisingly, rein tension in the halt was higher following a transition from the trot (16 N) than the walk (9 N) which reflects the difference in rein tension between the trot versus the walk. The median rein tension before (13.4 N) was higher than at halt (11.5 N), regardless of the gait before. In a study including 22 horses and three riders, a mean rein tension of 15 N and 13 N were applied on the left rein and the right rein, respectively, in order to halt the horse from walk but with between-horse variation from 2–43 N [16]. The differences and similarities for these figures may be explained by the difference in study design, regarding e.g., horse or rider experience, but can also be explained by the differences in the outtake of the relevant sequences that were studied (categorization of events was briefly addressed in [16]). In our halt model, a substantial portion of the (random) variation was accounted for by the riders, and horses within riders.

If riders are using the principles of negative reinforcement correctly [13], rein tension should decrease after the horse has performed a transition or halt. Equestrian manuals suggest that a transition should be accomplished through multiple half-halts with releases in-between [12]. In the current study, a release may be anticipated during the transitions and during the halts, but it was not evident in the rein tension curves. Within the transitions, rein tension is usually in-between the values for the gaits before and after, thus seemingly dependent on the gaits. The lowest least square means for rein tension during the transitions was at trot-to-walk type 1 with 12.8 N (Table 4). Thus, the decrease of rein tension during the downward transitions could either demonstrate a release of rein tension, or it could reflect the lower rein tension coupled with a slower gait. Theoretically, the rein tension during the non-moving halt could have been very light. Perhaps, the quite substantial median rein tension (11.5 N) at halt reflects the fact that the rider maintains the basic contact to keep the horse’s attention in preparation for the next transition rather than relaxing on long reins [10,22]. It is also possible that the horse contributed to this tension to some extent. In agreement with the rein tension at halt reported here, when riding a horse simulator at halt, a graph of rein tension indicates values in the range of 10–15 N [29].

A large proportion of the model variation was associated with the riders and horses within riders (the stage of transition model (b) and rein tension in halt model (c)) and the boxplots (Appendix A) support that riders differ substantially in the rein tension they use during the transitions. For example, rider 7 uses high median rein tension for the transitions (the boxplots) and for the median gait-specific rein tension. However, this rider also varies in the amount of rein tension applied for the transitions of the same transition category between horses, which is also found for rider 8. Figure 1 and Figure 2 exemplify both within-horse similarities and differences. Appendix A further demonstrates the number of transitions analysed per category and horse. The number of transitions per horse in the respective transition categories varied, which made the comparisons unbalanced. The individual patterns may be affected by factors such as the horse and rider level of education. The temperament and breed may also have influence as those factors have been shown to be associated with responsiveness to rein tension signaling when applied from the ground [30].

This was an observational study (from an epidemiological point of view). The data reflect the situation for the horse trained at home by a professional rider. All transitions were counted and none could be discarded because of less correct execution or equipment failure. Some transitions were difficult to see using the view from one camera and this also precluded a detailed analysis of horse behaviour. It was also not attempted to register the rider’s actions. To study rider (hand) movement and rein tension simultaneously could be informative, especially if aiming at studying the release of the reins.

There are several possible ways and perspectives from which our data could have been analyzed, and there is a multitude of factors that might influence rein tension during transitions. The factors such as rider, horse, time in session and ROM of nose angles were analysed for their association with rein tension at the transition. Further analysis focused on the rein tension time-sequence data, because rein tension in a particular sequence was predominately predicted by rein tension in the preceding sequence of the transition. However, the transitions that occurred very close in time (Figure 1) made it difficult to analyse whether sequences of rein tension long before (e.g., 4–5 s before a transition) would affect the rein tension just before or at the transition. For example, the rein tension at the first transition in this pair (Figure 1) is likely to have an effect on rein tension at the second transition in the pair and this was not analysed. While many comparisons were performed, only the median rein tension was analysed (some minima and maxima can be found in Appendix A), and perhaps this variable is not the best candidate if studying for example the release.

The transitions were coded from 25 Hz film and not from the 128 Hz head and croup signals. In future studies, video and kinematic data may be used in combination to delineate the transitions more precisely, including the continuous registration of stride length and stride frequency in preparing for and performing the transition. The placement of the head sensor did not allow a more exact determination of the actual nose angles [7] and only ROM was considered in the analysis. No power calculation was conducted for this pilot study. The number of transitions of each type, some in smaller numbers, and the variability between the riders and horses for many variables, suggest that a larger dataset may be needed to obtain a more efficient analysis. During the current classification of transitions, published principles were adhered to, also for the transitions (those between walk and canter) not previously classified in the literature [4,5]. There was also a difference between the coding of halts and transitions. During a transition was defined as when the horse changed between gaits, but halts were coded when the horse had stopped. Future studies may examine the during-the-transition-to-the-halt.

## 5. Conclusions

The results presented here indicate that changes in rein tension within a transition can largely be explained from differences in the head acceleration pattern between the respective gaits, with generally higher values at faster gaits. However, rein tension 1 s before the transition was generally higher than rein tension for the same gait during the session as a whole. Overall, tension was higher in the right rein. The finding that increased head motion in the period of 1 s preceding the transition had a significant effect on rein tension may be an indication that the horse is actively resisting the action of the bit. This suggests that riders should be diligent in seeking compliance from the horse. The large differences in rein tension between the horses and riders suggests that it is possible for some riders to make the horse perform transitions with a lighter contact, and that this is a feasible goal during training.

## Figures and Tables

**Figure 1 animals-09-00712-f001:**
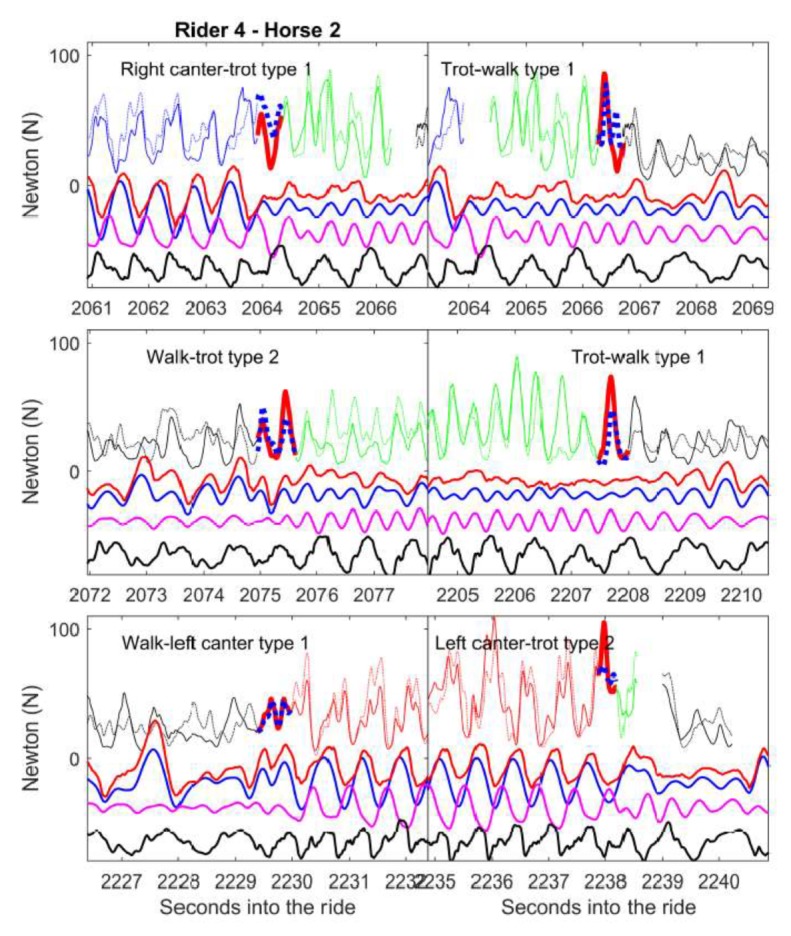
Examples of rein tension signals during six successive transitions in one horse. On each graph the upper curves show left and right rein tension (N) before, during and after the transition (horse 2/rider 4). The left rein is solid and the right dotted. The bolded part represents the actual transition (red/blue left/right rein). Black represents walk, green is trot and red/blue are left and right canter. Only one transition is featured per panel and rein data gaps represent other transitions, halts or unidentified gaits. The curves in the lower part of the graphs are dimensionless traces for the poll (blue) and croup (magenta) heights, croup roll (black, positive values represent downward motion of the left hip) and nose angle (red, positive forwards/nose out).

**Figure 2 animals-09-00712-f002:**
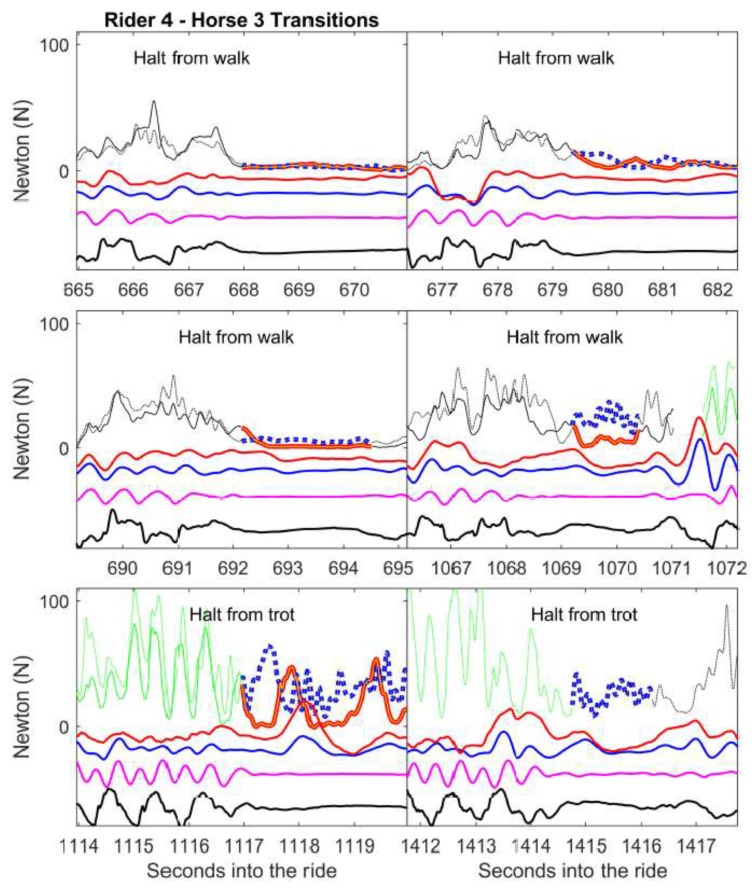
Examples of rein tension signals in transitions to halt. Within the six graphs (data from horse 3/rider 4), the upper curves show left (solid line) and right (dotted line) rein tension (N) before and during halts. The bolded part represents the halt/standstill (red/blue left/right rein). Black represents walk, and green is trot. Rein data gaps represent other transitions, halts or unidentified gaits. The curves in the lower part of the graphs represent dimensionless traces for the poll (blue) and croup (magenta) heights, croup roll (black, positive values represent downward motion of the left hip) and nose angle (red, positive forwards/nose out). Data are missing for the left rein in the last graph.

**Figure 3 animals-09-00712-f003:**
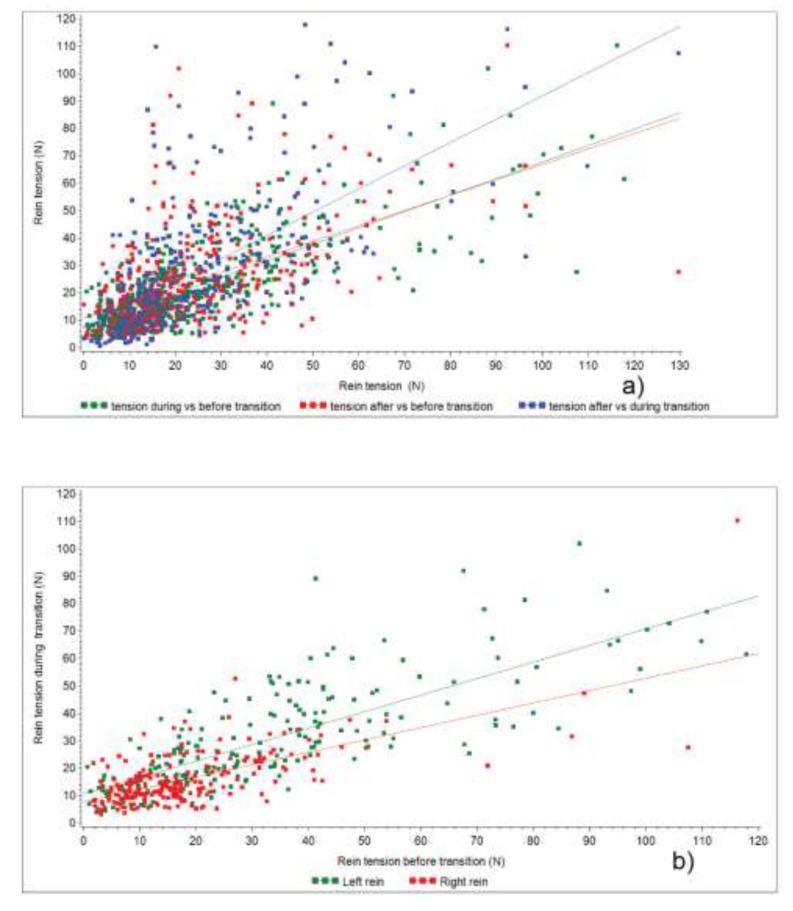
Median rein tension: (**a**) for left and right reins averaged, before, during and after transitions, and (**b**) median left and right rein tension before and during downwards transitions and upwards transitions. The fitted lines go with the symbols of the same colour, *n* = 427 in both plots.

**Table 1 animals-09-00712-t001:** The frequency of transition (*n* = 496) categories and types, and transitions to halt (*n* = 101) with right rein tension data (there was partial data loss for the left rein, Appendix A).

Transition	*n*	%	Transition	*n*	%
Category			Type		
Walk-trot	150	30	Walk-trot-type1	136	27
			Walk-trot-type2	14	3
Trot-walk	144	29	Trot-walk-type1	136	27
			Trot-walk-type2	8	2
Trot-canter	72	15	Trot-canter-type1	47	9
			Trot-canter-type2	25	5
Canter-trot	91	18	Canter-trot-type1	71	14
			Canter-trot-type2	19	4
			Canter-trot-other	1	0
Walk-canter	31	6	Walk-canter-type1	31	6
Canter-walk	8	2	Canter-walk-type1	7	1
			Canter-walk-type2	1	0
Halts					
Halt from walk	63	57			
Halt from trot	21	19			
Halt from rein back	14	13			
Other halts ^1^	13	12			

^1^ Other halts includes 2 with unidentified gait, 10 halts that were not seen from the camera angle and one from piaffe.

**Table 2 animals-09-00712-t002:** Model estimates with standard errors (SEs) from the exploratory a) mixed models where *p* < 0.10.

Outcome	Independent Variables	Estimate	SE	*p*-Value	
				Wald	Type III
Logarithm	Intercept	2.780	0.176	<0.0001	
of right	ROM ^1^ nose angle before transition	0.005	0.002	0.03	0.03
median rein tension	Intercept	1.469	0.085	<0.0001	
*n* = 496	Median left/right rein tension 1 s before	0.080	0.0048	<0.0001	<0.0001
	Square Median left/right rein tension 1 s before	−0.001	0.0001	<0.0001	<0.0001
	Intercept	2.679	0.214	<0.0001	
	Percentage in session	0.004	0.002	0.05	0.05
Average of	Intercept	2.732	0.146	<0.0001	
logarithm of left	ROM nose angle before transition	0.005	0.002	0.02	0.02
and right	Intercept	1.469	0.085	<0.0001	
median rein tension	Median left/right rein tension 1 s before	0.080	0.005	<0.0001	<0.0001
*n* = 427	Square median left/right rein tension 1 s before	−0.001	0.000	<0.0001	<0.0001

^1^ ROM—range of motion.

**Table 3 animals-09-00712-t003:** Least square means estimates (Est) with standard errors (SEs) for the stage of transition model (model b). The same letters within columns show significant differences (*p* < 0.05). There were only 1 transition between canter and walk of type 2, and 1 transition between canter and trot of undetermined type (other), hence comparisons involving these types have been omitted. *n* = 2769 observations based on 496 transitions.

Stage of Transition	Type	Est	SE	BT ^1^	Differences	
					Between Stages within Category/Type	Between Opposite Gait Pairs
Before	Walk-trot-type1	1.87	0.08	12.30	b	NE ^2^
During	*n* = 136	1.89	0.08	12.85	a	
After		1.99	0.08	15.62	ab	NE
Before	Walk-trot-type2	1.86	0.11	11.96	ac	NE
During	*n* = 14	1.92	0.11	13.69	bc	
After		2.05	0.11	17.81	ab	NE
Before	Trot-walk-type1	2.22	0.07	24.08	bc	NE
During	*n* = 136	2.09	0.07	19.03	ab	
After		1.95	0.07	14.43	ac	NE
Before	Trot-walk-type2	2.11	0.13	19.84		NE
During	*n* = 8	1.99	0.13	15.77		
After		1.96	0.13	14.63		NE
Before	Trot-canter-type1	2.15	0.09	21.53	b	NE
During	*n* = 47	2.18	0.09	22.54	a	a
After		2.24	0.09	25.33	ab	NE
Before	Trot-canter-type2	2.06	0.09	18.08		NE
During	*n* = 25	2.06	0.09	18.14		b
After		2.16	0.09	21.65		NE
Before	Canter-trot-type1	2.40	0.08	33.29	b	NE
During	*n* = 71	2.39	0.08	32.45	a	a
After		2.32	0.08	29.01	ab	NE
Before	Canter-trot-type2	2.41	0.10	33.99		NE
During	*n* = 19	2.50	0.10	39.19	a	b
After		2.38	0.10	32.35	a	NE
Before	Walk-canter-type1	2.00	0.09	16.16	a	NE
During	*n* = 31	2.02	0.09	16.54		c
After		2.23	0.09	24.64	a	NE
Before	Canter-walk-type 1	2.54	0.14	18.42	b	NE
During	*n* = 7	2.65	0.14	19.18	a	c
After		2.39	0.14	17.28	ab	NE
	Rein				Between reins	
	Left	2.18	0.07	22.66	a	
	Right	2.23	0.07	24.72	a	

^1^ BT—back transformed estimates; ^2^ NE—not evaluated.

**Table 4 animals-09-00712-t004:** Least square means estimates (Est) with standard errors (SEs) for rein tension before the transition model (model d). The outcome (transformed using the reciprocal inverse) is the difference per rein between the rein tension during 1 s before the transition and the median gait-and rein-specific rein tension during the session. Thus, the tension in a walk-trot type 1 transition was subtracted by the walk median rein tension (per rein) etc. *n* = 913 observations based on 491 transitions. Transition types represented with only 1 transition are omitted from the table.

Transition Type	Est	SE	BT 1 (N)
Walk-trot-type1	−0.00946	0.00012	5.71
Walk-trot-type2	−0.00948	0.00026	5.49
Trot-walk-type1	−0.00929	0.00012	7.64
Trot-walk-type2	−0.00972	0.00034	2.88
Trot-canter-type1	−0.00957	0.00017	4.49
Trot-canter-type2	−0.00995	0.00021	0.50
Canter-trot-type1	−0.00891	0.00014	12.23
Canter-trot-type2	−0.00892	0.00025	12.11
Canter-walk-type1	−0.00900	0.00036	11.11
eWalk-canter-type1	−0.00910	0.00019	9.89

^1^ BT—back transformed estimates.

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
