# Peer review of "Rein Tension in Transitions and Halts during Equestrian Dressage Training"

_animals, 2019, doi:10.3390/ani9100712_

Round 1

Reviewer 1 Report

As someone who is interested in Rein Tension I found this paper very interesting. This paper is also of particular value given its clear practical application.

I have made a number of comments on the attached manuscript. 

Three main aspects are:

1. the use of one vs two tailed hypotheses

2. clarity of the graphical information provided

3. the real need to clarify/correct text around learning theory, operant conditioning and especially reinforcement.

I hope that these comments are useful and assist with future versions. 

Best wishes

Author Response

Peer 1

Open Review

English language and style

( ) Extensive editing of English language and style required
(x) Moderate English changes required
( ) English language and style are fine/minor spell check required
( ) I don't feel qualified to judge about the English language and style

ANSWER: We have edited the manuscript in many places, independently as well as through the help of the reviewers.

Yes

Can be improved

Must be improved

Not applicable

Does the introduction provide sufficient background and include all relevant references?

( )

(x)

( )

( )

Is the research design appropriate?

(x)

( )

( )

( )

Are the methods adequately described?

(x)

( )

( )

( )

Are the results clearly presented?

(x)

( )

( )

( )

Are the conclusions supported by the results?

(x)

( )

( )

( )

Comments and Suggestions for Authors

ANSWER: We have improved the introduction in line with both reviewers’ comments and made additional language improvements for clarity.

As someone who is interested in Rein Tension I found this paper very interesting. This paper is also of particular value given its clear practical application.

ANSWER: Thank you.

I have made a number of comments on the attached manuscript. 

ANSWER: We have payed attention to all of these comments and amended all, in most cases directly as you suggested, and other occasions with further editing. Mentioning a few issues: we have added on rider handedness (and some reasoning around this) and we have added on surfaces and that horses were alone in the arena. We have not added on arena size- arenas were normal (20m*60-80m) to large sized. We added on bitting and training level. There was a mistake on how long the horses had been trained- so for this subset a mean is appropriate (median is also close) and now given as requested.

Three main aspects are:

the use of one vs two tailed hypotheses

ANSWER: Most statisticians agree that two-tailed hypotheses should in general, i.e. almost always, be used and certainly in this case as there is a biological possibility for changes in two directions. We have rephrased the wording of the hypotheses to some extent to make our reasoning fit better with that we used two-tailed tests. (We have also made minor adjustments in the discussion regarding our presumptions in 2 places).

clarity of the graphical information provided

ANSWER: For figure 3 we have changed the colours. We have (again) made sure the graphs have the correct resolution as deemed by the journal’s instructions and we mention ‘this’ concern to the editor so the end result hopefully becomes optimal.

the real need to clarify/correct text around learning theory, operant conditioning and especially reinforcement.

ANSWER: We have reviewed our statements and tried to improve the wording (in introduction and the discussion). Especially we had used terminology wrongly at one place in the discussion- which is now corrected.

Reviewer 2 Report

This is an excellent study which has addressed an aspect of riding (rein tension in transitions) which has to date not been fully considered in rein tension studies but which are an essential on modern riding technique. I would like to applaud the authors for undertaking this research as I feel the results contribute to the increasing evidence base which underpins practical horse riding / training. The study does apply an observational approach and whilst a more standardised experimental set up would have reduced limitations in data collection and facilitated more direct comparison between horse and rider dyads, I am confident the approach applied is valid and reliable, and personally I feel that taking a pragmatic approach using real-world data within this preliminary study will support its wider dissemination across equestrian stakeholders. I have made some minor suggestions for revisions below.

Simple summary – this provides a well written and easy to follow summary of the study

Lines 18-19: it would be useful to include the level of training / what level horses were working or competing at help readers understand their experience

Abstract:

Clear summary of the study provided

Line 29: suggest amending to ‘This study aimed to evaluate rein tension …’

Line 32: again here I think including briefly the level of horses experience would be beneficial

Lines 38-41: is there capacity here (although appreciate word count restrictions) to postulate why these differences occurred

Generally I would like some reference to P values within the abstract but feel their omission is appropriate here given the multiple models summarised.

Introduction:

The introduction outlines the role of transitions within riding well to the reader and will help those less familiar with equestrianism to understand the dynamic relationship between gaits – aids and rein tension, setting up the study well.

Line 69: it would be worthwhile to define or describe what a half-halt is and the role it plays for the less familiar reader

Line 85: suggest amending to ‘Our aim was to study…’

Line 87: there is an erroneous space between croup, and before

Line 89: please change to ‘hypothesized’ to retain past tense

Line 90 and 91: please amend ‘are’ to ‘would be’ for both uses

Line 92: suggest inserting ‘also’ between ‘would’ and ‘be’

Materials and methods:

Very comprehensive and justified methods included.

Line 100: please insert where further information can be found – I would suggest amending to ‘can be found in Eiserio et al. [16]’ to create a better flow in the sentence

Line 100: did you ask riders about their handedness? would be good to include this information if you have it or to state laterality preference of riders was not attained here

Line 105: suggest adding in ‘yard’ after ‘stable’ to clarify this was at a venue not in a stable!

Line 116: it would be beneficial to also include the frames per second the video camera recorded at

Line 123: please can you clarify if the normal flatwork session varied between riders / horses or was standardised, if the former as I believe then it would be useful to provide a summary of the average duration of flatwork sessions

Line 153: it would be useful to define what you considered ‘short reins’ to be here

Line 158: for descriptive median readings, the interquartile range should really be provided as well

Line 172: please make it clearer that this sentence refers to model a – as I read initially as the start of a list of points within the sentence

Line 192: odd space to remove between ‘model’ and ‘was’

Results:

Congratulations, you have presented a complex and large quantity of data in a manner which has a logical flow and is clearly communicated to your reader.

Line 262-266: it would be worthwhile to present one or two headlines from these data here to lead into the descriptions of what the supplementary information presents and to justify including in the main body of your manuscript

Lines 317-319: I would remove the parentheses form this sentence

Line 322: suggest adding in ’significant’ after remained to make a more complete sentence

Discussion:

Comprehensive discussion of the results provided. I particularly liked the application of data to transition sequencing and application to practice (i.e. riding) - this will support wider understanding of your results across a range of equestrian stakeholders. The authors have also demonstrated a strong awareness of the limitations within their data and considered the potential impact of these.

Lines 338 – 340: I would suggest level of training could also be influential on transition quality and maybe should be included here too?

Line 449: additional space requires removal between ‘analysing’ and ‘the’

Lines 448-456: it would be worthwhile to reiterate here that rider handedness was not assessed and to maybe add to final sentence that future research in this field should include record laterality of both horses and riders

Lines 475-480: could another potential explanation for the rein tension recorded at halt be the impact of the horse as rein tension measures the cumulative effect of horse and rider, and if the rider releases the rein / halts perhaps the horse’s contribution is being represented here?

Line 491: I would suggest level of training would also be a factor that would affect responsiveness

Line 506: start of this sentence is a little confusing to follow, it may work better as ‘For example’

Line 508: suggest replacing ‘done’ with ‘performed’

Conclusions:

Clear and appropriate conclusions provided.

Supplementary files support interpretation of the main manuscript and are clear and well presented.

Author Response

Peer 2

Review

English language and style

( ) Extensive editing of English language and style required
( ) Moderate English changes required
(x) English language and style are fine/minor spell check required
( ) I don't feel qualified to judge about the English language and style

ANSWER: We have edited the manuscript in several places, independently as well as through the help of the reviewers.

Yes

Can be improved

Must be improved

Not applicable

Does the introduction provide sufficient background and include all relevant references?

(x)

( )

( )

( )

Is the research design appropriate?

(x)

( )

( )

( )

Are the methods adequately described?

(x)

( )

( )

( )

Are the results clearly presented?

(x)

( )

( )

( )

Are the conclusions supported by the results?

(x)

( )

( )

( )

Comments and Suggestions for Authors

This is an excellent study which has addressed an aspect of riding (rein tension in transitions) which has to date not been fully considered in rein tension studies but which are an essential on modern riding technique. I would like to applaud the authors for undertaking this research as I feel the results contribute to the increasing evidence base which underpins practical horse riding / training. The study does apply an observational approach and whilst a more standardised experimental set up would have reduced limitations in data collection and facilitated more direct comparison between horse and rider dyads, I am confident the approach applied is valid and reliable, and personally I feel that taking a pragmatic approach using real-world data within this preliminary study will support its wider dissemination across equestrian stakeholders. I have made some minor suggestions for revisions below.

ANSWER: Thank you very much for these very encouraging words.

Simple summary – this provides a well written and easy to follow summary of the study

ANSWER: Thank you.

Lines 18-19: it would be useful to include the level of training / what level horses were working or competing at help readers understand their experience.

ANSWER: This information is now included in the simple summary. I have also added this to the body of the manuscript.

Abstract:

Clear summary of the study provided

ANSWER: Thank you.

Line 29: suggest amending to ‘This study aimed to evaluate rein tension …’

ANSWER: Amended as suggested.

Line 32: again here I think including briefly the level of horses experience would be beneficial

ANSWER: This information is now included in the abstract.

Lines 38-41: is there capacity here (although appreciate word count restrictions) to postulate why these differences occurred

ANSWER: We have used well over 200 words, so I think it is not possible to add explanations. Though we have remarked the relation between rein tension and gait- which adds a link between the result and the concluding part of the abstract (in line with the suggestion above).

Generally I would like some reference to P values within the abstract but feel their omission is appropriate here given the multiple models summarised.

ANSWER: Thank you. (I do agree that p-values should in general be included.)

Introduction:

The introduction outlines the role of transitions within riding well to the reader and will help those less familiar with equestrianism to understand the dynamic relationship between gaits – aids and rein tension, setting up the study well.

ANSWER: Thank you.

Line 69: it would be worthwhile to define or describe what a half-halt is and the role it plays for the less familiar reader

ANSWER: A good suggestion which we have addressed with a full sentence (and ref).

Line 85: suggest amending to ‘Our aim was to study…’

ANSWER: This part is rephrased based on reviewer 1’s suggestion.

Line 87: there is an erroneous space between croup, and before

ANSWER: Thanks for spotting.

Line 89: please change to ‘hypothesized’ to retain past tense

ANSWER: Thank you- amended.

Line 90 and 91: please amend ‘are’ to ‘would be’ for both uses

ANSWER: Thank you- amended.

Line 92: suggest inserting ‘also’ between ‘would’ and ‘be’

ANSWER: During reformulation of the hypotheses based on reviewer I’s suggestion this additional word was not needed.

Materials and methods:

Very comprehensive and justified methods included.

ANSWER: Thank you.

Line 100: please insert where further information can be found – I would suggest amending to ‘can be found in Eiserio et al. [16]’ to create a better flow in the sentence

ANSWER: Amended as suggested.

Line 100: did you ask riders about their handedness? would be good to include this information if you have it or to state laterality preference of riders was not attained here

ANSWER: Thanks for spotting this omission. Yes we also asked the riders about their handedness (not using a specific protocol at the time) and one was left handed and the rest right handed. We have now included this information.

Line 105: suggest adding in ‘yard’ after ‘stable’ to clarify this was at a venue not in a stable!

ANSWER: Inserted.

Line 116: it would be beneficial to also include the frames per second the video camera recorded at

ANSWER: This information is already included in the paragraph starting – ‘A second IMU’…

Line 123: please can you clarify if the normal flatwork session varied between riders / horses or was standardised, if the former as I believe then it would be useful to provide a summary of the average duration of flatwork sessions

ANSWER: We have inserted this information.

Line 153: it would be useful to define what you considered ‘short reins’ to be here

ANSWER: We have now provided a definition on the first occasion that we mention this. Thank you.

Line 158: for descriptive median readings, the interquartile range should really be provided as well

ANSWER: We have provided mean, SD, median and 5 and 95% percentiles in supplement S3 for all variables. This includes minimum and maximum rein tension during transitions and halts. We hope the reviewer find this information sufficient- but we can of course add to those columns if needed.

Line 172: please make it clearer that this sentence refers to model a – as I read initially as the start of a list of points within the sentence

ANSWER: Reworded slightly to avoid ambiguity.

Line 192: odd space to remove between ‘model’ and ‘was’

ANSWER: Thanks – amended.

Results:

Congratulations, you have presented a complex and large quantity of data in a manner which has a logical flow and is clearly communicated to your reader.

ANSWER: Thank you.

Line 262-266: it would be worthwhile to present one or two headlines from these data here to lead into the descriptions of what the supplementary information presents and to justify including in the main body of your manuscript

ANSWER: Good suggestion- added to this paragraph.

Lines 317-319: I would remove the parentheses form this sentence

ANSWER: Good suggestion- done.

Line 322: suggest adding in ’significant’ after remained to make a more complete sentence

ANSWER: Good suggestion- done.

Discussion:

Comprehensive discussion of the results provided. I particularly liked the application of data to transition sequencing and application to practice (i.e. riding) - this will support wider understanding of your results across a range of equestrian stakeholders. The authors have also demonstrated a strong awareness of the limitations within their data and considered the potential impact of these.

ANSWER: Thank you very much.

Lines 338 – 340: I would suggest level of training could also be influential on transition quality and maybe should be included here too?

ANSWER: Some rewording made in this section.

Line 449: additional space requires removal between ‘analysing’ and ‘the’

ANSWER: Amended.

Lines 448-456: it would be worthwhile to reiterate here that rider handedness was not assessed and to maybe add to final sentence that future research in this field should include record laterality of both horses and riders

ANSWER: Good suggestion. Updated with information on rider handedness added, while not analysed.

Lines 475-480: could another potential explanation for the rein tension recorded at halt be the impact of the horse as rein tension measures the cumulative effect of horse and rider, and if the rider releases the rein / halts perhaps the horse’s contribution is being represented here?

ANSWER: We have addressed this with an additional sentence.

Line 491: I would suggest level of training would also be a factor that would affect responsiveness

ANSWER: We agree and this has been added.

Line 506: start of this sentence is a little confusing to follow, it may work better as ‘For example’

ANSWER: Amended as suggested.

Line 508: suggest replacing ‘done’ with ‘performed’

ANSWER: Amended as suggested.

Conclusions:

Clear and appropriate conclusions provided.

ANSWER: Thank you.

Supplementary files support interpretation of the main manuscript and are clear and well presented.

ANSWER: Thank you.